

# Effects of fire disturbance on species and functional compositions vary with tree sizes in a tropical dry forest

Kanokporn Kaewsong[1], Chia-Hao Chang-Yang[2],
Sarayudh Bunyavejchewin[3], Ekaphan Kraichak[4], Jie Yang[5],
Zhenhua Sun[5], Caicai Zhang[6], Wenfei Li[7], Luxiang Lin[5] and
I-Fang Sun[1]

[1] Department of Natural Resources and Environmental Studies, College of Environmental Studies, National Dong Hwa University, Hualien, Taiwan
[2] Department of Biological Sciences, National Sun Yat-sen University, Kaohsiung, Taiwan
[3] CTFS-ForestGEO, Smithsonian Institution, Bangkok, Thailand
[4] Department of Botany, Faculty of Science, Kasetsart University, Bangkok, Thailand
[5] CAS Key Laboratory of Tropical Forest Ecology, Xishuangbanna Tropical Botanical Garden, Chinese Academy of Sciences, Yunnan, China
[6] Institute of Eastern-Himalaya Biodiversity Research, Dali University, Dali, Yunnan, China
[7] School of Ecology and Environmental Science, Yunnan University, Kunming, China

Corresponding author
Kanokporn Kaewsong,
kaewsong_k@hotmail.com

## ABSTRACT

**Background:** Disturbances are crucial in determining forest biodiversity, dynamics, and ecosystem functions. Surface fire is a significant disturbance in tropical forests, but research on the effect of surface fire on structuring species and functional composition in a community through time remains scarce. Using a 20-year dataset of tree demography in a seasonal evergreen tropical forest in Thailand, we specifically addressed two essential questions: (1) What is the pattern of temporal turnover in species and functional composition in a community with frequent fire disturbance? (2) How did the temporal turnover vary with tree size?

**Methods:** We analyzed species compositional and functional temporal turnovers in four different tree size classes among five tree censuses. We quantified species turnover by calculating Bray-Curtis dissimilarity, and investigated its underlying mechanisms by comparing pairwise dissimilarity of functional traits with simulations from null models. If fire disturbances contribute more to a stochastic process, the functional composition would display a random pattern. However, if they contribute more towards a deterministic process, the functional composition should reveal a non-random pattern.

**Results:** Over 20 years (1994–2014), we observed changes in species composition, whereas functional composition remained relatively stable. The temporal turnover patterns of species and functional compositions varied with tree sizes. In particular, temporal functional turnover shifted very little for large trees, suggesting that changes in species composition of larger trees are contributed by species with similar functional traits through time. The temporal functional composition turnovers of smaller trees (DBH ≤ 5 cm) were mostly at random. We detected a higher functional turnover than expected by null models in some quadrats throughout the 50-ha study plot, and their observed turnover varied with diameter classes.

**Conclusions:** Species compositional changes were caused by changes in the abundance of species with similar functional traits through time. Temporal functional turnover in small trees was random in most quadrats, suggesting that the recruits came from the equal proportions of surviving trees and new individuals of fast-growing species, which increased rapidly after fires. On the other hand, functional composition in big trees was more likely determined by surviving trees which maintained higher functional similarities than small trees through time. Fire disturbance is important for ecosystem functions, as changing forest fire frequency may alter forest turnover, particularly in functional composition in the new recruits of this forest.

# INTRODUCTION

Many studies have used a functional trait-based approach to understand the processes of community assembly and mechanisms that underlie the shifts in species composition (*Bernard-Verdier et al., 2012*; *Swenson et al., 2012*; *Purschke et al., 2013*; *Letten, Keith & Tozer, 2014*). Species' responses to the environment are mediated by their functional traits (*Sandel & Low, 2019*; *Wang & Wan, 2021*; *Guerin et al., 2022*), which affect species performance such as growth and survival (*Geber & Griffen, 2003*; *Violle et al., 2007*; *Reich, 2014*). Optimal functional traits of tree species vary along environmental gradients. For example, tropical trees in higher elevations tend to have thicker leaves and lower specific leaf areas than those in lower elevations. These leaf traits assist trees to tolerant in harsh environments, such as limited soil water condition (*Xu, Tomlinson & Li, 2019*). For these reasons, spatial and temporal variation in environmental conditions may lead to changes in species and functional composition.

Mechanisms underlying community dynamics involve stochastic and deterministic processes (*Swenson et al., 2012*). The relative importance of these two processes can be determined by evaluating temporal turnover in species and functional composition (*Swenson et al., 2012*). With only stochastic processes, such as ecological drift or dispersal limitation (*Hubbell, 2001*; *Chave, 2004*), a community is formed by a set of species that randomly enter or leave the community. Therefore, the temporal turnover in species composition is not related to changes in abiotic or biotic conditions, such as drought, predation pressure, and competition. As a consequence, the changes in functional composition would display a random pattern (*Swenson et al., 2012*). On the other hand, deterministic processes involve interactions among species and environment. As plant functional traits mediate species' responses to abiotic or biotic conditions, species turnover may be associated with species functional traits which are favored in certain environments (*Kraft, Valencia & Ackerly, 2008*; *Cornwell & Ackerly, 2009*). In this case, a community

would exhibit non-random turnovers in functional composition, either due to environmental filtering or biotic and abiotic factors.

Disturbances in tropical forests are critical driving forces of forest turnover, with consequent changes in species (*Denslow, 1980*; *Turner, Dale & Everham, 1997*; *Otterstrom, Schwartz & Velázquez-Rocha, 2006*; *Chazdon et al., 2007*) and functional composition (*Michalski, Nishi & Peres, 2007*; *Vanderwel, Coomes & Purves, 2013*; *Marra et al., 2018*). Shifts in species composition can alter ecosystem function (*Diaz & Cabido, 1997*; *Cardinale et al., 2011*), such as primary productivity and long-term carbon storage (*Chapin, Matson & Mooney, 2002*). In seasonal dry tropical forests in Southeast Asia, fire disturbance is a key element in maintaining forest structure and species diversity because it not only leads to death of trees but also provides opportunities for new recruits to establish (*Baker & Bunyavejchewin, 2017*). Fire events in tropical dry forests are usually low-intensity surface fires with short flame lengths and often occur during the dry season (*Stott, 1988*; *Baker, Bunyavejchewin & Robinson, 2008*; *Bunyavejchewin, Baker & Davies, 2011*). Surface fires usually cause high tree mortality rates for small-sized trees (*Slik & Eichhorn, 2003*; *Baker, Bunyavejchewin & Robinson, 2008*), open up space for recolonization, and boost seed germination of some tree species (*Fenner, 2000*; *Knox & Clarke, 2006*; *Otterstrom, Schwartz & Velázquez-Rocha, 2006*). In addition, loss of aboveground vegetation due to fire reduces potential neighborhood competition for surviving individuals. Fast-growing or pioneer species with functional traits like high leaf nutrient contents (phosphorus and nitrogen) may have a higher chance to establish after fire successfully (*Chai et al., 2016*). Accordingly, the post-fire community is likely composed of large surviving trees and new recruits from species with certain traits that adapt to the post-fire environment.

Many studies found fire disturbances could cause shifts in species composition (*Cochrane & Schulze, 1999*; *Gilliam & Platt, 1999*; *Cleary et al., 2006*). However, it is still unclear how fire influences turnover in functional composition, because fire can contribute to both stochastic and deterministic processes. When fire contributes to a stochastic process, it is likely to kill trees randomly or indiscriminately, regardless of species, size of trees, or functional traits. Consequently, functional turnover should exhibit random patterns. However, the role of the fire in shaping a community may be different, as a surface fire disturbance would kill mostly small trees, while a large canopy fire disturbance is more likely to kill almost all trees indiscriminately. Because of the non-random mortality, temporal turnover in functional composition should be non-random (*Swenson, Anglada-Cordero & Barone, 2011*; *Prado-Junior et al., 2016*). When larger trees have higher chance to survive after fire disturbances than smaller trees (*Slik & Eichhorn, 2003*; *Baker, Bunyavejchewin & Robinson, 2008*), we would expect the post-fire functional composition should be more similar to the pre-fire composition in large trees than in small trees. Even though there are studies that focused on the effect of fire disturbance on species turnover according to tree size variability in tropical forests (*Slik & Eichhorn, 2003*; *Baker, Bunyavejchewin & Robinson, 2008*), there is no study that investigates how functional turnover varies with size classes in relation to fire disturbances. Quantifying this functional turnover in different tree sizes could help to enhance our understanding of

mechanisms driving community dynamics. Such a study would also help us better predict future population dynamics and ecosystem functions.

To better understand the role of fires in structuring tropical dry forests, we quantified the effects of fire disturbance on species and functional composition of tree species using a 20-year tree demography dataset from a seasonal dry evergreen forest with recurrent surface fires. Specifically, we addressed the following questions: (1) What are the pattern of temporal turnover in species and functional composition? (2) How did the temporal turnover vary with tree size?

## MATERIALS AND METHODS

### Study site

The research was conducted in a seasonal dry evergreen tropical forest, the 50-ha Huai Kha Khaeng Forest Dynamics Plot (HKK FDP) from north-western Thailand (15°40′N, 90°10′E). The HKK FDP was established in 1990 to 1991. All trees with diameter at breast height (DBH, 1.3 m above ground) ≥1 cm were tagged with a unique number and identified to species (*Condit, 1998*; *Anderson-Teixeira et al., 2015*). Trees were measured and censused at a 5-year interval from 1994 to 2014. The plot contains about 300 tree species, most of which are evergreen. Dominant species include *Croton roxburghii* (Euphorbiaceae), *Dimocarpus longan* (Sapindaceae), and *Polyalthia viridis* (Annonaceae). The elevation of the plot ranges from 549 m to 638 m above sea level (*Bunyavejchewin et al., 1998*). The mean annual temperature is around 23.5 °C (*Bunyavejchewin et al., 2009*). Mean annual rainfall is 1,500 mm (*Baker & Bunyavejchewin, 2017*), with a 6-month dry season beginning in November (*Bunyavejchewin et al., 2003*). The major disturbances are low-intensity surface fires. There were three fire events recorded between 1990 and 2020 (*Baker, Bunyavejchewin & Robinson, 2008*; *Trouvé, Bunyavejchewin & Baker, 2020*), and fire occurred approximately once in a decade. The 1992 fire burned the south-east corner of the 50-ha plot (2 years before the first full census in 1994), the 1998 fire burned through the entire area of the plot (*Bunyavejchewin et al., 2009*), and the 2005 fire burned only some parts mostly around the border and very slightly in the middle of the plot (*Trouvé, Bunyavejchewin & Baker, 2020*). The HKK FDP had no records of human disturbances (*Bunyavejchewin et al., 1998*).

### Functional trait measurement

We collected functional trait samples of the 49 most abundant species (Table S1) from June to September 2016. We collected leaf samples from trees across four DBH size classes: (1) ≤5 cm, (2) >5 cm to ≤10 cm, (3) >10 cm to ≤20 cm, and (4) >20 cm. We sampled at least three individual trees in each diameter class of each species. We collected three leaves and ten leaflets from the species with simple and compound leaves, respectively, except for of *C. roxburghii*. For *C. roxburghii*, only two individual trees from the smallest size class were sampled. All leaf samples were collected from trees inside the 50-ha plot. We measured leaf chlorophyll content (Chlo, SPAD) from each individual leaf by using a hand-held 'SPAD-502 Chl meter' (Minolta Camera Co., Osaka, Japan). We measured leaf area (LA, cm$^2$), leaf dry matter content (LDMC, mg g$^{-1}$), leaf thickness (LT, mm), and

specific leaf area (SLA, $cm^2 g^{-1}$) of each leaf. We calculated the mean values of each species and each size class from these individual leaves. For leaf chemical traits, we took mixed-sample leaves as represented for size class mean values to measure leaf total carbon concentration (LCC, $g kg^{-1}$), leaf total nitrogen concentration (LNC, $g kg^{-1}$), leaf total phosphorus concentration (LPC, $g kg^{-1}$), and leaf total potassium concentration (LKC, $g kg^{-1}$). All leaf functional traits were measured according to *Pérez-Harguindeguy et al. (2013)*. We collected five wood samples from individuals outside the plot for each tree species and measured wood density following the ForestGEO protocol (www.forestgeo.si.edu/protocols). We calculated the mean wood densities of each species. Note that due to the limitation of collecting wood samples across size classes, we assigned the wood densities of all four size classes of each species the same values as takenfrom the mean wood density of that species.

## Quantifying temporal turnover in species and functional composition

We compared species and functional composition of 1,250 20 m × 20 m quadrats between different censuses. We used the 1994 census as a baseline and calculated the temporal turnover of the following time intervals: 1994–1999, 1994–2004, 1994–2009, and 1994–2014. The 20 m × 20 m is an appropriate scale to investigate forest turnover for tropical forests as recommended by *Swenson et al. (2006)* and *Swenson et al. (2012)*. We quantified species turnover for the community from the 49 most abundant species and for each diameter class using an abundance-based metric, the Bray-Curtis dissimilarity ('vegdist' function in the package 'vegan' in R). To calculate temporal turnover in functional traits, we performed the log10-transformation of the functional trait values to achieve normality prior to the analysis. We then standardized all functional traits by subtracting the mean and divided by the standard deviation. We calculated the pairwise dissimilarity of the functional traits using the Euclidean distance-based measurement ('dist' function in the package 'stats') of each 20 m × 20 m quadrat to represent the temporal turnover in functional composition in each quadrat. We did the same analysis for temporal functional turnover in each of the four tree diameter classes (1–5, 5–10, 10–20, >20 cm). The formula of pairwise dissimilarity calculation is described in *Swenson et al. (2012)*. This gives:

$$D_{PW} = f_A \sum_{i=1}^{S_A} f_i \overline{\delta_{ib}} + f_B \sum_{j=1}^{S_B} f_j \overline{\delta_{ja}}$$

where $S_A$ and $S_B$ are the total number of tree species at time A and B in the community, respectively; $f_i$ is the relative abundance of species $i$ at time A, and $f_j$ is the relative abundance of species $j$ at time B in the community. The mean pairwise functional distance between species $i$ at time A and all species at time B in the community is represented by $\overline{\delta_{ib}}$ and the mean pairwise functional distance between species j at time B and all species at time A in the community is represented by $\overline{\delta_{ja}}$. $f_A$ is the total number of trees at time A divided by the total number of trees at time A and B, and $f_B$ is the total number of trees at time B divided by the total number of trees at time A and B.

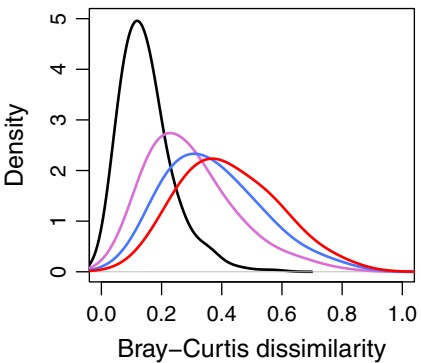

**Figure 1 Kernel density estimates of Bray-Curtis dissimilarity for temporal community-wide turnover in species composition varying in time intervals.** The dissimilarity values were calculated at 1,250 20 m × 20 m quadrats in the 50-ha Huai Kha Khaeng Forest Dynamics Plot, Thailand. Line colors represent time intervals in 1994–1999, 1994–2004, 1994–2009, and 1994–2014 in black, purple, blue, and red respectively. Increasing dissimilarity values indicate increasing differences in species composition through time. Variations in kernel density highlight that the shifts in species composition tend to increase with the increasing time.

### Null models for temporal turnover in functional trait composition

We used a null model approach to test if the observed pairwise functional dissimilarity value was significantly different from random. We constructed null distributions of temporal turnover in functional trait combinations for each quadrat using 9,999 randomizations. In each randomization, species names were randomly drawn from all studied species (49 species) for each quadrat. We calculated the pairwise functional dissimilarity of each quadrat using the randomized data. We then compared the observed value with the null distribution and estimated the quantile score (rank) of the observed pairwise dissimilarity value. The quantile scores of observed functional pairwise dissimilarity were significantly lower or higher than expected when the quantile scores of the observed values fell within the bottom 2.5th or the top 97.5th percent quantiles, respectively. This also implies that the temporal turnovers in functional composition were more similar or dissimilar than expected by chance.

## RESULTS

### Temporal species turnover

The Bray-Curtis (BC) dissimilarities tended to increase with the time intervals (1994–1999, 1994–2004, 1994–2009, and 1994–2014). This indicates that the species composition of most quadrats changed over time. On average the BC dissimilarities between 1994 and 1999 were the lowest in comparison to other intervals (Fig. 1 but see additional information in Fig. S1). The distribution of this interval had the highest peak compared with other time intervals.

The temporal turnover in species composition varied across diameter classes (Fig. 2 and Fig. S2). The BC dissimilarities tended to increase with the time interval between censuses for all diameter classes, and the shift was most evident in trees at the smallest diameter class (DBH ≤ 5 cm). In other words, the differences in BC dissimilarities tended to decrease

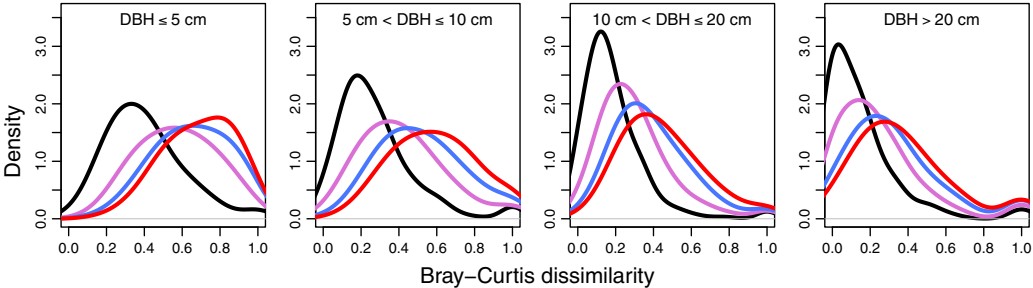

**Figure 2 Kernel density estimates of Bray-Curtis dissimilarity for temporal turnover in species composition for four different time intervals according to the tree diameter size classes.** The dissimilarity values were calculated at 1,250 20 m × 20 m quadrats in the 50-ha Huai Kha Khaeng Forest Dynamics Plot, Thailand. Line colors represent time intervals in 1994–1999, 1994–2004, 1994–2009, and 1994–2014 in black, purple, blue, and red respectively. We show results for all four DBH size classes as right-hand panels (large trees) highlight the much lower species turnover comparing to left-hand panels (small trees).

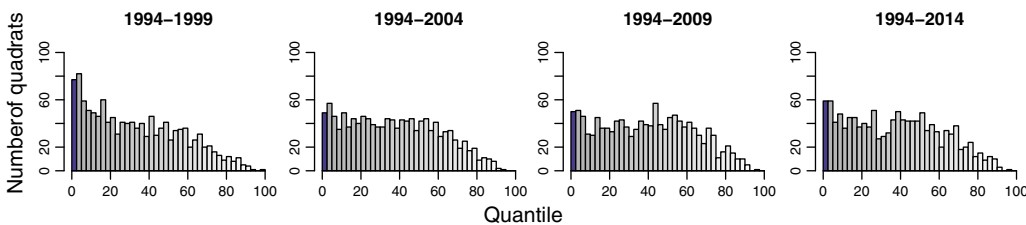

**Figure 3 Histograms show quantile scores of functional pairwise dissimilarity for community-wide turnover in functional composition for different time intervals at 1,250 20 m × 20 m quadrats in the 50-ha Huai Kha Khaeng Forest Dynamics Plot, Thailand.** Quantile scores represent values of significantly smaller or higher than expected by null models at 2.5 (blue bars) or 97.5 (yellow bars) respectively, at the 0.05 level. The other bars in greyscale represent the values between 2.5 and 97.5 (low values in grey and high values in white). Each bar corresponds to an interval of 2.5.

with tree sizes. Furthermore, the average values were the lowest in 1994–1999 in all diameter classes. Trees in the largest size class (DBH > 20 cm) had the lowest compositional turnover between 1994 and 1999.

### Temporal functional turnover

The temporal functional turnover did not increase with time intervals (Fig. 3 and Fig. S3). Overall, the quantile scores of pairwise functional dissimilarity were not significantly different from the null models in most quadrats. Within 1,250 quadrats, 77, 49, 50, and 59 quadrats exhibited functional turnover significantly different from random in 1994–1999, 1994–2004, 1994–2009, and 1994–2014, respectively, (Fig. 3). In addition, these quadrats are primarily distributed around the middle of the 50-ha study plot (Fig. 4).

While the overall temporal turnover in functional composition remained unchanged, the functional turnover decreased with tree size (Fig. 5 and Fig. S4). The number of quadrats that had functional turnover significantly lower than expected by random increased with tree size. In most quadrats, pairwise dissimilarities of saplings and treelets (DBH ≤ 5 cm and 5 cm < DBH ≤ 10) were not significantly different from null models.

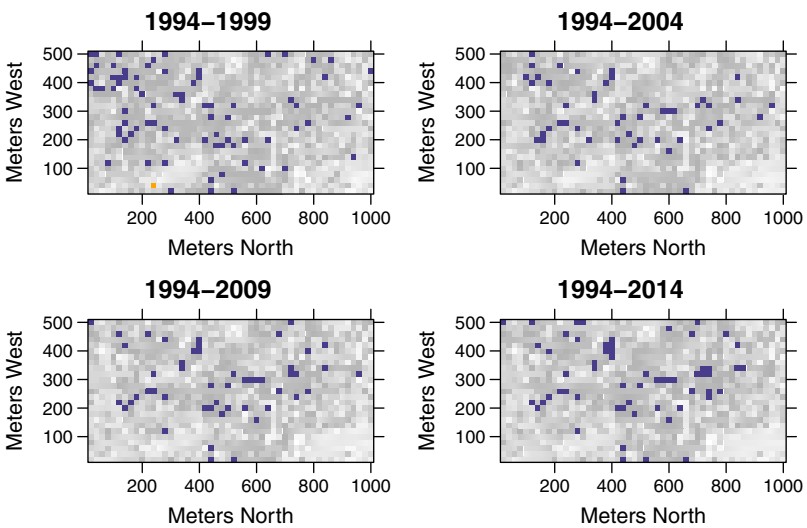

**Figure 4 Spatial distributions of quantile scores of functional pairwise dissimilarity for community-wide turnover in functional composition for different time intervals at 1,250 20 m × 20 m quadrats in the 50-ha Huai Kha Khaeng Forest Dynamics Plot, Thailand.** Quantile scores represent values of significantly smaller or higher than expected by null models at 2.5 (blue squares) or 97.5 (yellow squares) respectively, at the 0.05 level. The other squares in greyscale represent the values between 2.5 and 97.5 (low values in grey and high values in white).

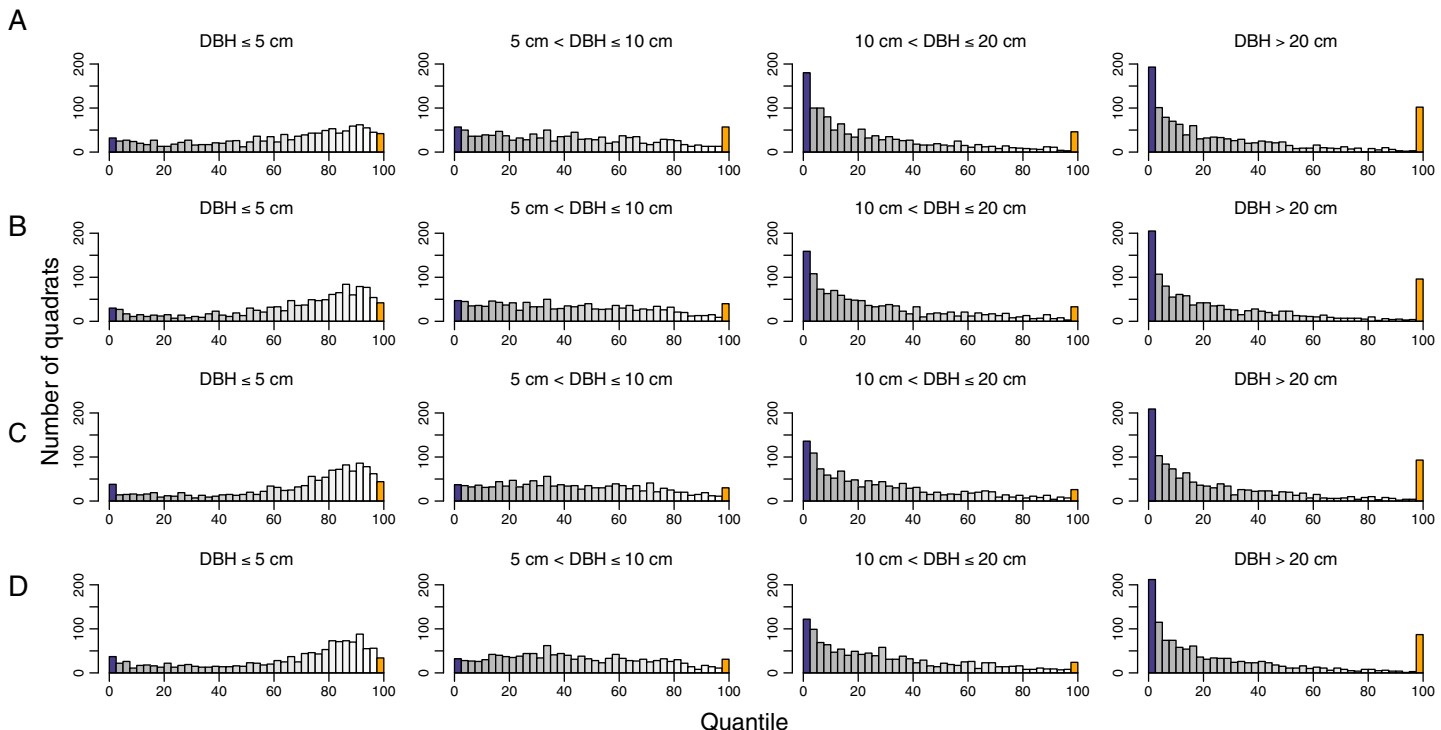

**Figure 5 Histograms show quantile scores of functional pairwise dissimilarity for turnover in functional composition for different time intervals and diameter classes at 1,250 20 m × 20 m quadrats in the 50-ha Huai Kha Khaeng Forest Dynamics Plot, Thailand.** (A) 1994–1999, (B) 1994–2004, (C) 1994–2009, and (D) 1994–2014. Quantile scores represent values of significantly smaller or higher than expected by null models at 2.5 (blue bars) or 97.5 (yellow bars) respectively, at the 0.05 level. The other bars in greyscale represent the values between 2.5 and 97.5 (low values in grey and high values in white). Each bar corresponds to an interval of 2.5.

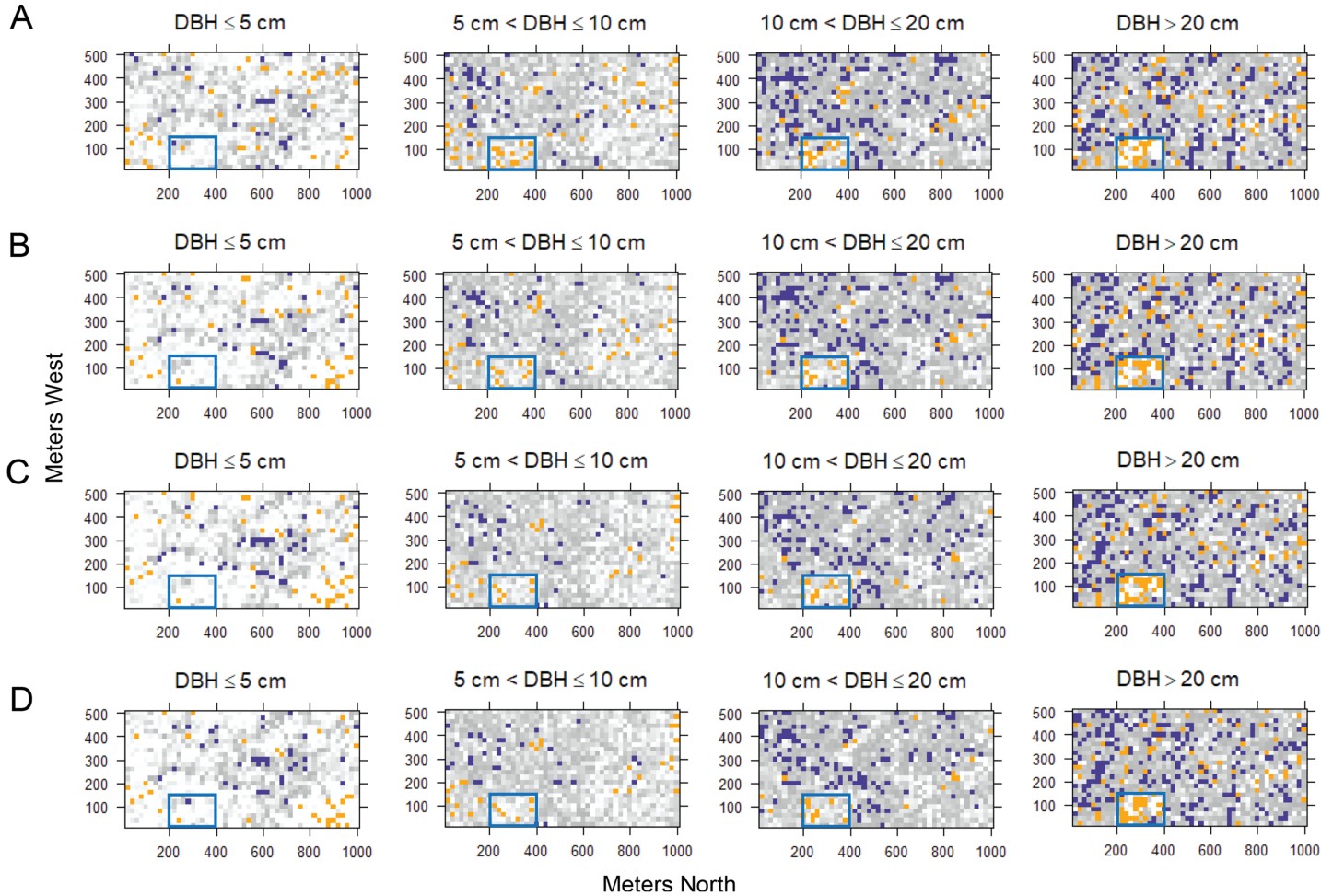

**Figure 6 Spatial distributions of quantile scores of functional pairwise dissimilarity for turnover in functional composition for different time intervals and diameter classes at 1,250 20 m × 20 m quadrats in the 50-ha Huai Kha Khaeng Forest Dynamics Plot, Thailand.** (A) 1994–1999, (B) 1994–2004, (C) 1994–2009, and (D) 1994–2014. Quantile scores represent values of significantly smaller or higher than expected by null models at 2.5 (blue squares) or 97.5 (yellow squares) respectively, at the 0.05 level. The other squares in greyscale represent the values between 2.5 and 97.5 (low values in grey and high values in white). The blue rectangle represents area with high elevations (around x = 200 to 400 m and y = 0 to 150 m).

The quadrats with pairwise dissimilarities significantly higher than null models were distributed differently within the HKK FDP among size classes (Fig. 6). That is, in the smallest size class the quadrats with significantly higher functional turnover distributed around the southwest corner of the plot, whereas in the largest size class the quadrats with significantly higher functional turnover primarily occurred at the high elevations (Fig. 6).

## DISCUSSION

### Effect of fire on turnover in species composition

Our results showed that the abundance-based species compositional changes tended to increase with the time intervals in HKK in the past two decades (Fig. 1 and Fig. S1). One possible reason is that fire events occurred prior to both 1994 and 1999 tree censuses, and the recovery time from fires was similar for both censuses (Table S2). In 1992, a fire

burned some parts of the study site, and in 1998 a fire burned around 1,500 km$^2$ of the surrounding area and thoroughly across the 50-ha study plot (*Baker, Bunyavejchewin & Robinson, 2008*). There are only two years apart between the 1992 fire and the 1994 census, and one year apart between the 1998 fire and the 1999 census. As surface fires killed mostly small individuals (*Baker, Bunyavejchewin & Robinson, 2008*), species composition 1 or 2 years after a fire was probably composed of surviving trees and recruits of fast-growing species. The number of trees in the 50-ha plot declined from 78,923 to 74,347 individuals between 1994 and 1999, probably due to the 1998 fire which killed a large number of individuals. Since then, only one fire occurred in 2005 and the abundance in the 50-ha plot increased to more than 127,000 trees in 2014.

Even though surface fires have been observed to reduce the number of trees, the abundance of many species can quickly recover with a few years (*Cochrane & Schulze, 1999*; *Baker, Bunyavejchewin & Robinson, 2008*). This is because the reduction in tree density opens up more space for recruitment. In addition, a decrease in tree density may weaken density dependent effect due to competition or natural enemies, which may lead to higher seedling survival rate (*Augspurger, 1984*). Fast-growing species, such as *C. roxburghii*, had high mortality during fire events but can quickly recover after fire disturbances (*Baker, Bunyavejchewin & Robinson, 2008*). Unlike other slow-growing species, these species may benefit from the reduction in tree density caused by fires. Abundances of *C. roxburghii* increased greatly from 8,896 in 1994 to 33,990 in 2014. On the other hand, *Baccaurea ramiflora*, a thin bark species (*Baker & Bunyavejchewin, 2006*) that fires greatly affect its mortality (*Baker, Bunyavejchewin & Robinson, 2008*), has slightly decreased in abundance from 2,567 in 1994 to 2,048 in 2014.

The patterns of temporal turnover in species composition across size classes (Fig. 2 and Fig. S2) were similar to the overall pattern (Fig. 1 and Fig. S1). Our results particularly showed a strong species turnover pattern in small trees through time. This finding highlights that species composition in big trees was more stable than in small trees over time. The effect of surface fire may contribute to these patterns. Big trees are more effective in preventing the heat of fires from entering their living tissues than small trees (*Gutsell & Johnson, 1996*; *Brando et al., 2012*). While fires kill a huge number of small trees (*Cochrane & Schulze, 1999*; *Baker, Bunyavejchewin & Robinson, 2008*; *Brando et al., 2012*), new recruits of diverse species enter the post-fire community. According to our results, species turnover is lower in larger trees (DBH > 5 cm), perhaps due to low mortality rates and limited recruitment in large size classes. Even though a previous study in this forest has reported that fires could effectively kill large trees of some common species such as *C. roxburghii*, *B. ramiflora*, and *P. viridis* (*Baker, Bunyavejchewin & Robinson, 2008*), the mortality of these large trees was relatively low. For example, the 1998 fire burned through the study plot, and the mortality of the larger *C. roxburghii* trees was 34 times lower than that of the smaller trees in the 1999 census. We also found that the largest-sized trees had the lowest compositional turnover between 1994 and 1999 (Fig. 2). The result may reveal the relatively minor effects of 1992 and 1998 fires on species composition in big trees. Those surviving big trees that persist in 1994 and 1999 might contribute to the composition similarities in 1994 and 1999.

## Effects of fire on turnover in functional composition

Our results showed that functional turnover in many quadrats was not different from the expectations of stochastic processes (Fig. 3 and Fig. S3). The results indicate that the temporal shifts in functional composition in the whole plot are predominantly random. In other words, the changes in tree species composition in this forest are mostly random through time in terms of functional trait combinations. However, the functional turnover was significantly lower than null expectations in some quadrats (Figs. 3 and 4). This implies that the observed temporal turnover patterns in these quadrats were a result of deterministic processes, such as environmental filtering, that favor species with optimal functional traits in that environment (*Kraft, Valencia & Ackerly, 2008*; *Cornwell & Ackerly, 2009*; *Jung et al., 2010*). In our case, high canopies of dipterocarps in the flat areas around the middle of the plot create a low-light condition that is suitable for shade-tolerant species. These species usually have dense wood, providing these trees with resistance from physical damages under the shade (*van Gelder, Poorter & Sterck, 2006*). After these shade-tolerant trees die, new recruits under the same dipterocarp canopies will be constrained by the light condition, resulting in another set of shade-tolerant species with dense woods. Consequently, functional composition in this area will remain similar through time.

Furthermore, we found functional composition resulting from species with highly functional similarities was more stable in 1994–1999. After the 1992 and 1998 fire disturbances, the recolonizing species possessed similar functional traits to those before the fire. Consequently, the functional turnover in 1994–1999 was lower than in other years (Fig. 3). Another reason could be that, based on species turnover results, we found species compositions were similar between 1994 and 1999. Species turnover during this time was possibly mainly determined by trees that survived both the 1992 and 1998 fires.

As species composition varied through time, the functional composition remained unchanged. One possible reason for this apparent decoupling between species and functional turnover is that species that died during this study period were replaced by species with similar traits through some deterministic processes. Our findings are consistent with previous research in the 50-ha Barro Colorado Island Forest Dynamics Plot (BCI FDP) in Panama. Functional composition in BCI FDP was relatively stable through time, despite there being shifts in species composition (*Swenson et al., 2012*).

We further observed functional trait distributions of each trait over time in the community, which could help us better explain the temporal trends in functional composition. Only a few functional traits shifted through time. WD slightly decreased, while LPC increased with the time interval (Fig. 7). Accordingly, species that grow fast commonly have high values of LPC, LNC (*Chai et al., 2016*), SLA (*Poorter et al., 2008*; *Prado Júnior et al., 2015*), and low WD (*Enquist et al., 1999*; *Muller-Landau, 2004*). If fires kill many trees and create suitable habitats for recruits of fast-growing species, one would expect functional composition after fire disturbances to be contributed mainly by fast-growing species. Consistent with this expectation, we found that the dominant and fast-growing species *C. roxburghii*, which has low WD and high LPC, greatly increased in

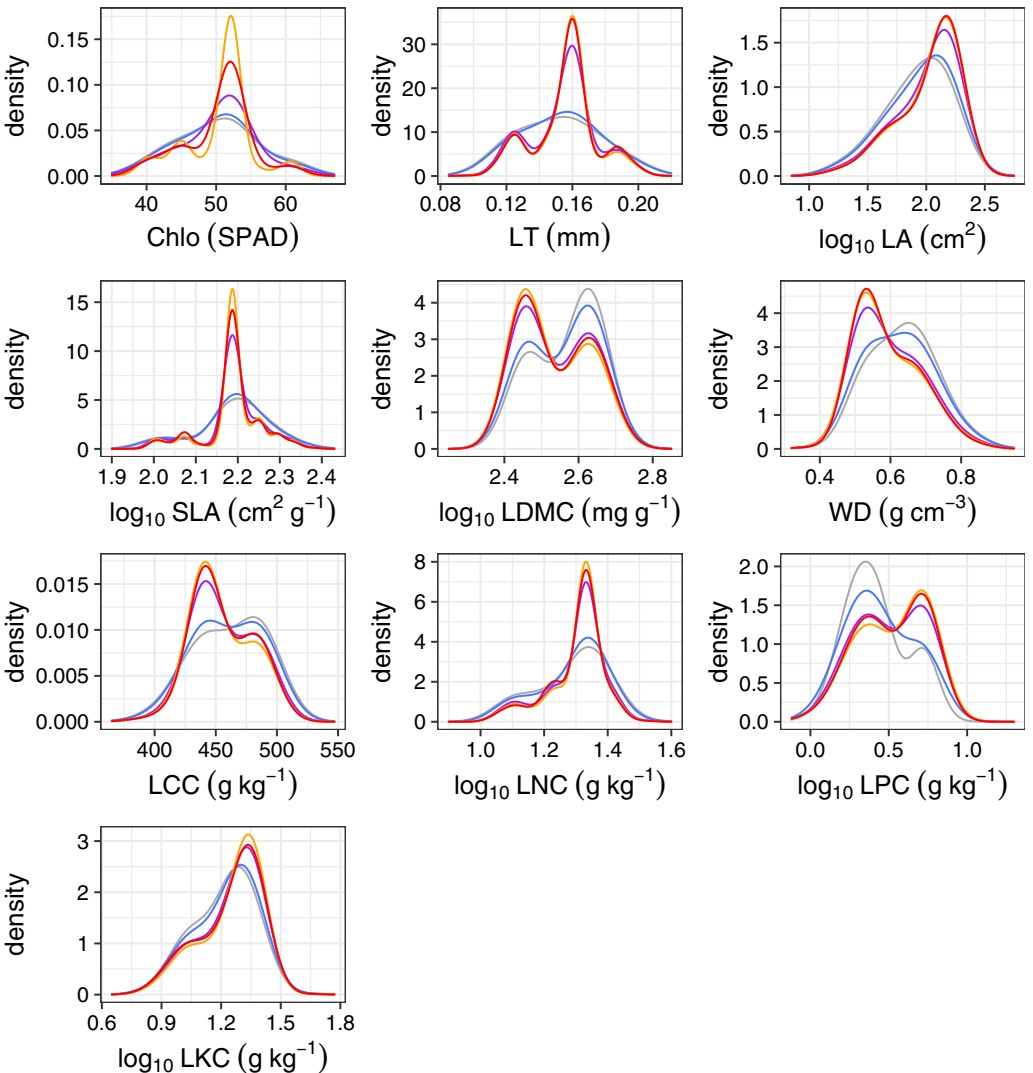

**Figure 7 Kernel density estimates of functional traits of all trees from 49 study species at the 50-ha Huai Kha Khaeng Forest Dynamics Plot, Thailand.** All trees of the same species are assigned to have the same trait values derived by species mean traits. Chlo, Chlorophyll content; LT, leaf thickness; LA, leaf area; SLA, specific leaf area; LDMC, leaf dry matter content; WD, wood density; LCC, leaf total carbon concentration; LNC, leaf total nitrogen concentration; LPC, leaf total phosphorus concentration; and LKC, leaf total potassium concentration. Line colors represent census years in 1994, 1999, 2004, 2009, and 2014 in grey, blue, purple, orange, and red respectively.

abundance over time. This implies that a prominent change in the abundance of this species may have caused the increase in LPC and decrease in WD. Other functional traits varied very little through time, suggesting that this forest is generally composed of species with similar functional traits over time. Thus, the compositional changes in species in this forest are mainly caused by shifts in the abundance of a dominant fast-growing species and species with similar functional traits over time.

  Temporal functional turnover varied with tree sizes and exhibited similar patterns in species turnover. We found that big trees (DBH > 10 cm) mostly showed significantly lower functional turnover than small trees (Fig. 5). The functional turnover in many

quadrats of big trees was significantly lower than expected by null models, indicating that the shifts in species composition of big trees were due to replacement by species with similar functional traits. In other words, most large trees that survived fires through time determined the functional composition after burning. The temporal functional turnover of small trees appeared to be more random than big trees. The composition of small trees may be equally contributed by recruits of fast-growing species and surviving trees. The areas which contained most quadrats with random patterns in functional turnover were distributed throughout the plot. Remarkably, high temporal functional turnover was observed in some quadrats which differed according to areas and varied among diameter size classes (Fig. 6).

Higher temporal functional turnovers than expected by null models mainly occurred in many quadrats around the southwest corner of HKK FDP for trees of the smallest size. Functional turnovers in these areas are possibly determined by the increase in *C. roxburghii* abundance. On the other hand, quadrats with significantly high functional turnover in big trees were distributed mainly at high elevations (Fig. 6). In fact, these areas are relatively dry and have a high percentage of rock with low soil moisture compared to lower elevations. There are few big trees in comparison to low elevations, and only few species, such as *Lagerstroemia tomentosa*, live specifically in these areas (*Bunyavejchewin et al., 2009*). Big trees of *L. tomentosa* disappeared in some quadrats while other species such as *C. roxburghii* and *Mallotus philippensis* established in these areas through time. Consequently, temporal functional turnover in these areas was perhaps driven by species with different functional traits.

Although surface fires could kill some large trees for some species (*Baker, Bunyavejchewin & Robinson, 2008*), other factors, such as drought, may also be important. Drought not only induces fire occurrences (*Cleary & Genner, 2004*; *Cochrane, 2009*) but also increases the mortality risk of large trees (*Bennett et al., 2015*). During the study period (1991–2014), several ENSO events were recorded (1991–1992 (*Harger, 1995*), 1993–1994, 1997–1998 (*Wooster, Perry & Zoumas, 2012*)). Droughts associated with these ENSO events may also contribute to the death of the big trees in HKK.

Although temporal turnover in functional composition changed very little, functional turnover varied greatly throughout the entire 50-ha plot (Figs. 4 and 6). This variation across quadrats might be affected by the spatial heterogeneity of fire intensity (*Trouvé, Bunyavejchewin & Baker, 2020*). Factors such as light conditions, soil and topographic variables at local scales might also contribute to the spatial variation in the functional turnover patterns. For instance, the soil moisture in the quadrats at low elevations was relatively high, influencing the fire occurrence and intensity (*Trouvé, Bunyavejchewin & Baker, 2020*), thus contributing to the temporal forest turnover in HKK.

## CONCLUSIONS

Overall, this study demonstrates that temporal shifts in species composition are not associated with functional compositional changes in a seasonal dry evergreen tropical forest. This suggests that the temporal changes in species composition were caused by species with similar functional traits. Both species compositional and functional turnover

varied with tree size. Surface fires might kill many small trees and, at the same time, promote dominant and fast-growing species that are well adapted to fire disturbances to increase after burning. Accordingly, functional similarities were lower in these small trees. In contrast, big trees that survived fires could persist and dominantly act as determinants of similar functional composition through time. Other environmental factors might also co-determine in these compositional changes through time.

The limitation of our study is that we could not include functional traits related to fire tolerance and survival strategies, such as bark thickness and resprouting ability. Further, we could not assess the effects of intraspecific (inter-individual) variation in trait values and fire regimes like fire intensity which may affect the results. Therefore, future works should include fire-related functional traits, intraspecific trait variation, and detailed fire information in the analysis.

## ACKNOWLEDGEMENTS

We thank the Smithsonian Institution's Center for Tropical Forest Science that supported tree census. We thank Dr. Sarayudh Bunyavejchewin's team, Ms. Milian Yang, Mr. Lang Ma, Mr. Yongzhen Shen, and Ms. Supparat Tongkok for their contributions to the collection of functional traits in the study site. We also thank Drs. Yiching Lin, Jyh-Min Chiang, Shih-Chieh Chang, and Yu-Yun Chen for their suggestions on data analysis.

### Funding

This study was funded by the Strategic Priority Research Program of the Chinese Academy of Science (XDB31000000), the Joint Fund of the National Natural Science Foundation of China-Yunnan Province (U1902203), and the Southeast Asia Biodiversity Research Institute, Chinese Academy of Sciences (151C53KYSB20200019). The funders had no role in study design, data collection and analysis, decision to publish, or preparation of the manuscript.

### Grant Disclosures

The following grant information was disclosed by the authors:
Strategic Priority Research Program of the Chinese Academy of Science: XDB31000000.
National Natural Science Foundation of China-Yunnan Province: U1902203.
Southeast Asia Biodiversity Research Institute, Chinese Academy of Sciences: 151C53KYSB20200019.

### Competing Interests

The authors declare that they have no competing interests.

### Author Contributions

- Kanokporn Kaewsong conceived and designed the experiments, performed the experiments, analyzed the data, prepared figures and/or tables, authored or reviewed drafts of the paper, and approved the final draft.
- Chia-Hao Chang-Yang conceived and designed the experiments, authored or reviewed drafts of the paper, and approved the final draft.
- Sarayudh Bunyavejchewin conceived and designed the experiments, authored or reviewed drafts of the paper, and approved the final draft.
- Ekaphan Kraichak conceived and designed the experiments, authored or reviewed drafts of the paper, and approved the final draft.
- Jie Yang performed the experiments, authored or reviewed drafts of the paper, and approved the final draft.
- Zhenhua Sun performed the experiments, authored or reviewed drafts of the paper, and approved the final draft.
- Caicai Zhang performed the experiments, authored or reviewed drafts of the paper, and approved the final draft.
- Wenfei Li performed the experiments, authored or reviewed drafts of the paper, and approved the final draft.
- Luxiang Lin conceived and designed the experiments, authored or reviewed drafts of the paper, and approved the final draft.
- I-Fang Sun conceived and designed the experiments, authored or reviewed drafts of the paper, and approved the final draft.

## Data Availability

The raw data are available in the Supplemental File.

## Supplemental Information

Supplemental information for this article can be found online at http://dx.doi.org/10.7717/peerj.13270#supplemental-information.

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
