# Peer review of "Effects of fire disturbance on species and functional compositions vary with tree sizes in a tropical dry forest"

_PeerJ, doi:10.7717/peerj.13270_

## Round 0.1 · original submission · Major Revisions

Please revise and improve the manuscript according to the comments of three reviewers.

Reviewer 1 ·

Basic reporting

The paper is clearly written and easy to understand, although there are minor grammatical English errors throughout.

The paper is well organized, has good literature review and sufficient background, and results and discussion stick to the questions posed in the introduction.

It is clear what the figures show and they are good quality, and relevant to the questions being explored in this analysis.

Experimental design

A 50 ha mapped plot with census of trees every 5 years provides an excellent basis for the study, with 1250 20 x 20 m quadrats in which species composition is followed.

This is original research on an important topic with respect to dry tropical forests. The research questions are well defined and important, namely how surface fire affect the species composition, size class distribution and functional trait composition of the forest over 20 years.

The methods are well explained and the way in which functional traits were determined in the field and used in the statistical analysis for the 49 species followed were very good.

The results are rigorous and because repeated observations over 20 years in the same forest after fire is uncommon in tropical dry forests, it does fill a knowledge gap.

Validity of the findings

The findings presented are valid. However, I don't think the data was fully analyzed. In addition to 5, 10, 15 and 20 year time periods, all starting in 1994, the study could also use the same data to do four 5-year periods within the total 20-year period of observation (1994-1999, 2000-2004, 2005-2009, 2010-2014). This is important because the 1994-1999 results (shape of curves in Figures 1 and 2) is much different than the curves for 1994-2004, 1994-2009, 1994-2014. To really understand what is going on, we should also see if the remaining 5-year periods after 1994-1999 also have curves shaped the same, or if the five-year periods look progressively more like the 1994-2014 curves. In other words, is 1994-1999 a typical 5-year period or is there progression in successive 5-year periods toward the shapes of the curves for the full 20-year period, so that the analysis mostly represents forest recovery from the 1998 fire? With this addition the conclusions would be stronger.

Conclusions are linked to the questions posed in the introduction. The data presented is adequate to verify the results and conclusions.

Additional comments

The role of fire is not integrated into the abstract--although it does mention that surface fires play an important role in dry tropical forests, the occurrence of fire during the 20 years of observation and its role in the results are not mentioned.

Reviewer 2 ·

Basic reporting

Comment 1: In community ecology, one important goal is to understand the processes of community assembly and the underlying mechanisms. Therefore, many processes have been proposed to explain the patterns of plant diversity, including stochastic and deterministic processes, which can be determined by evaluating temporal turnover in species and functional composition. Fire disturbance is an important driving force of forest turnover, affecting ecosystem functioning in tropical forests. The authors aimed to address two important questions: 1) what are the pattern of temporal turnover in species and functional composition in a community with frequent fire disturbance? And 2) how did the temporal turnover vary with tree size? The topic of this study is interesting and important. However, the statements are at time unclear, especially for the fire effects on temporal species and functional turnover for tree species.
Comment 2: The title is “How does surface fire disturbance affect temporal species and functional turnover for tree species in a tropical dry evergreen forest?”. However, I failed to find how fire (e.g. burn severity, burned area or fire return interval) affected temporal species and functional turnover for tree species. The authors did not test the direct effects of fire on post-fire temporal species and functional turnover. And in the conclusion, the author stated that they did not assess the effects of fire regimes like fire intensity. In my opinion, I recommend changing the title.
Comment 3: In lines 57-58, the authors took an example to support above statement “Optimal functional traits of tree species vary along environmental gradients”. The example is related to the trees that survive in high mountainous areas with low temperatures tend to have small leaves adapted to low temperature communities. I think it would be better to take an example which is related to how functional traits of tropical trees respond to environmental changes.
Comment 4: In general, fire is a random event in landscape-scale and is therefore difficult to predict. In the manuscript, the author stated that “fire can contribute to both stochastic and deterministic processes” and explained this statement from lines 98 to 107. From the statements, the authors explained how fire caused the stochastic process, but did not introduce the deterministic process clearly. The statement “while a high intensity canopy fire such large canopy fire is more likely to kill almost all trees indiscriminately” is confusing. Whether there is a mistake in the statement of “high intensity canopy fire such large canopy fire”. Furthermore, the authors stated that “because of the non-random mortality, temporal turnover in functional composition should be non-random”. Please provide related references for this. Although tree mortality is random with a high intensity fire, the post-fire recruitment process may be random as the seed dispersal is random from the unburned forest.
Comment 5: In the introduction, the authors stated the underlying effects of fire intensity on temporal species and functional turnover. However, they did not test the effects of fire intensity on temporal species and functional turnover. So the introduction seems to be redundant.

Experimental design

No comments

Validity of the findings

Comment 1: The author stated that “we quantified the effects of fire disturbance on species and functional composition of tree species”. Whereas in the conclusion, they stated “we could not assess the effects of intraspecific trait variation and fire regimes like fire intensity”. The statements seem to be contradictory.
Comment 2:Comment 7: It is hard for readers to see how the quadrats which exhibited functional turnover significantly different from random primarily distributed in the 50-ha study plot in the figure S1 due to the similar green colors. Same comments for figure S2.
Comment 3: In the discussion section, the two subheadings of “The effect of tree size on species temporal compositional changes” and “The effect of tree size on temporal functional turnover” discussed the effects of tree size on species temporal compositional changes and functional turnover, not related to fire effects. This also suggested that the title was not appropriate.

Additional comments

Comment 1: line 112 There was a mistake for “be help”. Maybe "helpful"?
Comment 2:line 228 and 230 There was a mistake for the word“high”. Maybe “higher”?

Reviewer 3 ·

Basic reporting

1)The references cited in the Introduction section should be updated and added. There are few cited references of recent five years in the manuscript. The newest references could better describe the novelty and make the readers who focus on related research field understand the importance of current research.

2) L211-212. How authors reached this conclusion? Please provide the detailed information here. This statement is not apparent in Fig.2.

3) Fig. S1 and S2 are critical for current research because authors referred these two figures so many times in the manuscript. I suggest that authors move these two figures into the manuscript instead of putting it in the supplementary materials.

4) Result section. Why the results with high DBH (>10cm) of Fig.4 are not presented in here? Besides, the results of Fig.4 are abnormal due to extremely high numbers of quadrats appeared in the second, third and fourth column. Can you explain or provide more detail?

Experimental design

no comment

Validity of the findings

5) The novelty and contribution should be clearly bolded in the discussion section. Therefore, the structure of Discussion needs to be modified to better answer two questions raised in the Introduction. I suggest that authors clearly summarize the results by dividing the discussion into three parts, which mainly discuss the turnover pattern (Fig.1 and 3), the effect of sizes (Fig.2 and 4) and changes of functional trait distribution (Fig.5).

Additional comments

1) L32. Should be “investigated”
2) L50. Missing words: “new recruits this forest” should be “new recruits of this forest”
3) L55. Should be “affect”
4) L65. Please add the reference.
5) The forms of Fig.2 and Fig.3 should be uniform. There should be one title of x-axis and y-axis.
6) The caption of figures should be adjusted. The names of Fig.1 and Fig,2, Fig.3 and Fig.4 are identical.

Annotated reviews are not available for download in order to protect the identity of reviewers who chose to remain anonymous.

---

## Round 0.2 · accepted · Accept

Compared with version 0, the current manuscript and attachments have been improved according to the comments of the reviewers, and the key questions have been answered and solved. I think this manuscript is acceptable.

Reviewer 1 ·

Basic reporting

Improvements made in response to previous review are very good

Experimental design

No comment

Validity of the findings

Revisions in response to previous review make it clear that findings have high validty.

Additional comments

Changes in response to my previous review comments seem well done. I have no additional comments at this time.